# Magnetic Core-Shell Nanoparticles Using Molecularly Imprinted Polymers for Zearalenone Determination

**DOI:** 10.3390/molecules27238166

**Published:** 2022-11-23

**Authors:** Luis Calahorra-Rio, Miriam Guadaño-Sánchez, Tamara Moya-Cavas, Javier Lucas Urraca

**Affiliations:** Department of Analytical Chemistry, Faculty of Chemistry, Complutense University of Madrid, Plaza Ciencias, 2, 28040 Madrid, Spain

**Keywords:** zearalenone, mycotoxins, molecular imprinting, magnetic nanoparticles, solid phase extraction

## Abstract

This paper describes the synthesis of novel molecularly imprinted magnetic nano-beads for the selective extraction (MISPE) of zearalenone mycotoxin in river and tap waters and further analysis by high-performance liquid chromatography (HPLC) with fluorescence detection (FLD). A semi-covalent imprinting approach was achieved for the synthesis of the molecularly imprinted polymers (MIP). The nanoparticles were prepared by covering the starting Fe_3_O_4_ material with a first layer of tetraethyl orthosilicate (TEOS) and then with a second layer using cyclododecyl 2-hydroxy-4-(3-triethoxysilylpropylcarbamoyloxy) benzoate. The last was used with a dual role, template and functional monomer after the extraction of the template molecule. The material was characterized by transmission electron microscopy (TEM), X-ray diffraction (XRD) and Fourier transform infrared spectroscopies (FT-IR). The solid phase extraction was optimized in all the steps: loading, washing and elution. The optimal conditions allowed the determination of zearalenone in trace levels of 12.5, 25 and 50 µg L^−1^ without significant differences between the fortified and found level concentrations.

## 1. Introduction

Mycotoxins are low molecular-weight natural toxic compounds, biologically classified as secondary metabolites, produced by some types of moulds or fungi species. When these compounds are ingested, inhaled, or absorbed through the skin, they can cause diverse health problems or even death in humans and several animal species [1,2]. Cereal crop contamination due to fungal infection is a well-known phenomenon that can happen either in the field or while the product is transported, storage or even processed. Moreover, even after being only at trace levels, mycotoxins represent a global problem because of their accumulation in foodstuffs. Their occurrence can have catastrophic economic consequences [3,4].

*Fusarium* species are one of the most widespread families of fungi. They can be found on almost all continents and are known to infest both pre- and post-harvested crops, resulting in the contamination of human foods and animal feed. The most important and most common mycotoxin-generating species in harvested fruits and vegetables are *F. culmorum*, *F. roseum* and *F. graminearum*; being the last the main primary producer of zearalenone (ZON) [5]. This non-steroidal toxin is found worldwide in the moderate temperature regions of Europe, Asia and America [6]. ZON presence has been reported in a large number of cereal crops, mainly in maize, barley, wheat, oats and sorghum and their derivate grain products such as bread, beers, and processed feeds. ZON is a very stable compound that maintains its structure throughout all the storage and milling stages and even during the industrial processing/cooking procedures [7,8]. Biologically, ZON is considered a powerful estrogenic compound, whose hormonal action exceeds that of most other naturally occurring non-steroidal estrogens, including soy and clover isoflavones [9]. This mycotoxin has been directly related, by several studies, with malfunction in the reproductive tract of laboratory and domestic animals. Some of the estrogenic alterations detected were: decreased fertility, increased fetal resorptions, and changes in the weight of endocrine glands and serum hormone levels [9,10]. Based on these data, the European Food Safety Authority (EFSA) established in 2014 in Europe the maximum tolerated levels of ZON for human consumption as: 20 μg kg^−1^ in food intended for babies and infants, 50 μg kg^−1^ in maize-based snacks and breakfast cereals, and 200 μg kg^−1^ in unprocessed maize and certain maize products [11]. Although there are currently no legislative maximum residual limits for ZON in waters, ZON can be present in waters along with other mycotoxins such as fumonisin B3, ochratoxin A and several types of aflatoxins [12].

Accurate and sensitive analytical methodologies had focused attention since this regulation was set, owing to the mandatory quantification of ZON; both to ensure consumer’s and livestock’s health and to develop the international markets [13]. Regarding the traditional methodology, considerable literature can be found, from use of thin-layer chromatography (TLC), gas chromatography coupled with mass spectrometry (GC–MS) or liquid chromatography using fluorescence or mass spectrometric detection to the use of enzyme-linked immunosorbent assays (ELISA) [14,15,16,17,18,19,20,21,22,23,24]. Determination of ZON in complex matrices usually requires clean-up and pre-concentration steps before the analysis; these prior steps are necessary to improve the method’s sensitivity and selectivity. Therefore, conventional methods rely on time-consuming, laborious sample pretreatment procedures and high-rise costs [25,26].

Molecularly imprinted polymers (MIPs) are robust and cost-efficient smart materials which have the valuable ability to recognize selected target molecules [27,28]. The selective affinity hotspots are created by a two-step process: (1) template molecules are imprinted into the polymeric matrices according to their shape, size and functional group distribution; (2) removal of the template molecule. Hence, extremely selective hollows are generated in the three-dimensional polymeric structure. During recent years, great attention has been focused on improving the synthesis conditions of MIPs. According to the approach used to prepare these MIPs, we can find up to three families [29]. First, the non-covalent imprinting approach: this procedure is based on relatively weak interactions such as electrostatic and hydrophobic interactions, hydrogen bonding or π-π bonding established between the template and structural polymer [30,31,32,33]. In second place, the covalent approach is based on covalent bond formation between the template molecule and the functional monomer before polymerization; once the polymer has been synthesized, the template molecule is extracted from the 3D-network disrupting the covalent bonds. Thus, the rebinding process takes place due to covalent bonds reformation [34]. Finally, a third way uses a combination of the above two: the semi-covalent approach. This strategy is based on the establishment of covalent bonds during the polymerization step, as explained for the covalent approach, but non-covalent interactions result in the rebinding process [35,36]. This approach offers the selectivity of the covalent approach and the versatility attributed to the non-covalent imprinted polymers.

Magnetic properties associated with magnetic nanoparticles have sparked interest during the last decades because of their properties, such as low costs, biocompatibility and large functionality [37,38]. These particles (particularly Fe_3_O_4_) offer optimum characteristics to prepare core-shell hybrids resulting in magnetic molecularly imprinted polymers (MMIPs) which combine the magnetic response of the core with the tailored selectivity of the MIP shell. This surface modification also improves the dispersity, selectivity and biocompatibility of the particles [39,40]. Nowadays, MMIPs are extensively used, due to their attractive properties and their easy and efficient separation, in fields such as separation methods [41], catalysis [42], bioscience [43] or environmental remediation [44].

Some works have been already published for the determination of ZON using MIPs [33,45,46]. Nevertheless, typical approaches (using non-covalent bonds) have been applied in the study of ZON in organic media, for recognition in solvents such as acetonitrile. Extraction of this mycotoxin from cereals is more effective using this kind of solvent. However, ZON can also be present in aqueous samples, although no reports have been presented on effective measurement. This work reports for the first time the synthesis of a semi-covalent molecular imprinted polymer using magnetic nanoparticles as a nucleus that allows the determination of zearalenone in aqueous samples.

## 2. Results and Discussion

### 2.1. Synthesis of Fe_3_O_4_@MIP

Traditionally, the solvothermal method has been used for the synthesis of magnetic nanoparticles based on Fe_3_O_4_, due to its ease of preparation. This method is based on the use of a teflon reactor for 24 h where the reaction mixture is maintained at high temperature (190 °C). Nevertheless, despite the simplicity of this method, the polydispersity in the size of the particles obtained is generally high, which leads to a much more heterogeneous coating in later steps.

In order to achieve a simple solvothermal procedure that allows obtaining a more homogeneous particle size, a new proposal was carried out using MW. Therefore, a comparative study of the particle size in the synthesis was achieved with the solvothermal method and particles synthesized by MW.

As shown in the transmission electron micrograph in Figure 1A, using the solvothermal method, spherical particles with an average diameter of 87 ± 27 nm were obtained. In the second approach, the synthesis was performed in a microwave reactor at 200 °C for 8 min, considerably shortening the synthesis time. Thus, magnetic nanoparticles were obtained with an average diameter similar to that obtained previously 85 ± 11 nm, but with a smaller size dispersion (Figure 1B). This new methodology allows the reduction of the time for synthesis, and reduced the polydispersity of the nanomaterial; all the later sets of nanoparticles were synthesized according to this method.

Regarding the coating of the magnetic nanoparticles with a TEOS (Fe_3_O_4_@SiO_2_), the size of the shell layer was also optimised. For this purpose, three different methods were used. With the first one (Method A), the agglomeration of the nanoparticles was favoured and the coating obtained was not homogeneous (Figure 2a). A reduction in the amount of TEOS was then used with an increment of the polymerization time up to 72 h, without modifying any other parameter (Method B). The shell formed around the nanoparticles was not homogeneous and the silica grow up in an irregular way on the Fe_3_O_4_ surface. Finally, maintaining the conditions of method B, the synthesis was repeated but in the presence of oleic acid (Method C). The presence of oleic acid contributes to stabilizing the individual Fe_3_O_4_ nanoparticles. With the latter method, spherical and individual nanoparticles with a shell of 20 ± 4 nm were obtained as it is shown in Figure 2c. Thus, this method was used for the subsequent coating of the molecular imprinted polymers.

To obtain the final Fe_3_O_4_@SiO_2_@MIP and in agreement with the literature [45], CDHB (Section 3.4.1) template was demonstrated to be a very good candidate for the imprinting effect of zearalenone (Figure 3). In this case cyclododecyl 2-hydroxy-4-(3-triethoxysilylpropylcarbamoyloxy) benzoate was used with a dual role in the synthesis. First, in the polymerization process it was used for creating the well defined cavities for the formation of the imprinting polymer. Second, after extraction of the template (Figure 4) using a mixture of dioxane:water (7:1 *v*/*v*), the hydrolysis process took place, and the silane derivative obtained acted as a monomer in the cavity for the zearalenone recognition. Instead a typical non-imprinted polymer (NIP) was used as a control polymer (CIP). A CIP is a polymer made in the same conditions that the MIP but using a different template molecule used for the MIP. The decision of using a CIP instead a conventional NIP (without template) was because the NIP shell could present a different morphology in absence of the template molecule. Thus, with the purpose of evaluating the non-specific interactions, a polymer made in the same conditions but using methyl 4-(3-triethoxysilylpropylcarbamoyloxy) benzoate as a template was used.

### 2.2. Morphology and Structure of the Core—Shell Nanostructures

#### 2.2.1. Fourier Transform Infrared Spectroscopy (FT-IR)

In the FTIR spectrum of the magnetite (Appendix A black line), the 558 cm^−1^ band was assigned to the Fe-O bond stress vibrations of the Fe_3_O_4_, being the most representative of magnetite nanoparticles. In the FTIR spectrum of the Fe_3_O_4_@SiO_2_@MIP (red line), the 3449 cm^−1^ and 1201 cm^−1^ bands were assigned to the tensile vibration of the O-H bonds due to the OH groups present in the SiO_2_ layer formed around the magnetic Fe_3_O_4_ particles. Finally, the 1631 cm^−1^ band was assigned to the C=O group tension band present in the carbamate and benzoate groups contained in the template molecules used in the covalent imprinting of MIP and CIP, respectively.

#### 2.2.2. Transmission Electron Microscopy

Transmission electron microscopy (TEM) images of the Fe_3_O_4_@SiO_2_ (Figure 2) and Fe_3_O_4_@SiO_2_@MIP (Figure 5) nanocomposites were performed to corroborate the formation of the polymer layer on SiO_2_. An increase in the thickness of the Fe_3_O_4_ magnetic nanoparticle coating corresponding to the SiO_2_ plus sol-gel polymer layer was observed (25 ± 4 nm). Thereby, one of the main objectives of this work could be achieved: creating a not too large polymer film to avoid analyte diffusion problems from the solution to the binding sites without obtaining aggregated particles.

#### 2.2.3. X-ray Powder Diffraction

Appendix A compares the XRD patterns of the magnetic cores before and after the polymerisation process, allowing confirmation of whether the polymerisation conditions altered the structure of the MNPs. The diffraction pattern of the MNPs before coating showed diffraction maxima indexed to 220, 311, 400, 422, 440 and 511 reflections, typical of the cubic inverse spinel structure of magnetite (JCPDS fiche no. 19-0629 for Fe_3_O_4_). The diffraction pattern of the nanocomposites Fe_3_O_4_@SiO_2_ and Fe_3_O_4_@SiO_2_@MIP (red and blue line respectively) showed diffraction maxima indexed to 220, 311, 400, 422, 440 and 511 reflections, as in the case of the uncoated MNPs. In general, most of the peaks have lower intensity for nanocomposites than for uncoated MNPs; this is because the peaks after the coatings are less defined and broader due to the presence of the polymer layer surrounding the nanoparticles.

Therefore, it can be concluded that neither the silica coating nor the formation of the polymer on the MNPs caused any phase change in the iron oxide NPs. The average size of the uncoated MNPs, the Fe_3_O_4_@SiO_2_ and the Fe_3_O_4_@SiO_2_@MIP crystals, determined from the Scherrer’s formula (D = Kλ/β cos θ) [47] were estimated to be 14.9; 14.7 and 15.0 nm, respectively. The crystal lattice parameter was not significantly altered by the coatings (0.834 ± 0.001; 0.835 ± 0.002 and 0.835 ± 0.001 nm Fe_3_O_4_, Fe_3_O_4_@SiO_2_ and Fe_3_O_4_@SiO_2_@MIP, respectively); moreover, it was similar to that of standard magnetite (a = 0.8396 nm).

#### 2.2.4. Microanalysis

Comparing the carbon percentage values shown in Appendix A for all samples, the presence of carbon was observed in the Fe_3_O_4_ magnetic cores and in the Fe_3_O_4_@SiO_2_ nanocomposites due to the presence of residues of the solvents used to carry out the corresponding syntheses. For the naocomposite with the polymeric layer, Fe_3_O_4_@SiO_2_@MIP/CIP, a large increase in the percentage of carbon was observed due to the presence of all the carbon present in the template molecule. Finally, in the hydrolysed MIP/CIP, a decrease in the amount of carbon was observed, demonstrating the effectiveness of the method used for the extraction of the template molecule.

#### 2.2.5. Thermogravimetric Analysis

As shown in Appendix A, for all three types of nanoparticles there was a weight loss up to 200 °C, attributable to the loss of water and other solvents that might be retained in the corresponding polymeric networks. From 200 °C to 750 °C the observed weight variation is attributed to the loss of organic matter in the individual particles. Finally, the remaining weight loss is attributed to the thermal resistance of the amorphous SiO_2_.

It can be observed that the weight loss of the Fe_3_O_4_@SiO_2_@MIP (black line) particles is higher than that of these particles after hydrolysis of the template molecule (red line), which in turn is higher than that of the Fe_3_O_4_@SiO_2_ particles (purple line). This behavior was expected since Fe_3_O_4_@SiO_2_@MIP nanoparticles were the ones that contained a higher amount of organic matter in their structure, which would be reduced after the hydrolysis of the template molecule. On the other hand, the weight loss for Fe_3_O_4_@SiO_2_@MIP was higher than for Fe_3_O_4_@SiO_2_ nanoparticles, since they were only coated with a TEOS layer. Moreover, it was observed that the weight loss for the latter nanoparticles was faster which could be explained considering that the bond formed between the two materials is weaker than the one formed after the synthesis of the second layer.

#### 2.2.6. Magnetization

The silica layer formed around the Fe_3_O_4_ could affect the magnetic properties of the ferromagnetic iron oxide nanoparticles after the polymerization process. The hysteresis loop for the initial nanoparticles (before polymerization) is typical of a superparamagnetic material: no coercivity or remanence is observed (Appendix A). Identical results were obtained after the polymerization process: The core-shell nanoparticles were as superparamagnetic as the original ones. Saturation magnetization was considerably lower (13 vs. 33 emu g^−1^ at room temperature). This was clearly the result of polymerization: only the core contributed to the magnetic signal but both the ferromagnetic core and the polymeric shell contributed to the mass, which resulted in decreased saturation magnetization.

### 2.3. Binding Isotherm

The binding properties and the homogeneity of the binding sites of the polymer were assessed by equilibrium analysis. A rebinding test was carried out at pH = 7.5, where higher differences were expected between the imprinted and the non-imprinted polymer. To perform equilibrium binding isotherms of ZON to MIP and CIP (Figure 6a), 1 mg of the polymer was incubated with mycotoxin concentrations in the range of 0.002–0.030 mM for 15 min. Thus, a rebinding test using ZON was achieved. The binding features of ZON to both the MIP and CIP polymers were accurately modelled to a Langmuir binary site model.

Unfortunately, among the typical fits used to fit these isotherm models (Freundlich, Langmuir…), none of them could be used to fit both isotherms simultaneously. As shown in Figure 6a, the shape of both is very different, which prevents a correct fitting following the same model for both. For this reason, it was decided to fit both curves to the same scatchard model where both the affinity constant and the total number of binding sites can be easily compared. The graph in Figure 6b represents the amount of ZON bound to the polymer (B, μmol g^−1^) versus the concentration of free toxin in the supernatant (F, µmol L^−1^) and the resulting fit originated two classes of union sites (Figure 6b,c). The differences in the isotherms of the evaluated polymers show that the binding capacity of the MIP is higher than that of the corresponding NIP. The binding constants over the concentration range evaluated (0.002–0.030 mM) for MIP class 1 and 2 sites were: (1) K_MIP,ZON_ = 314,620 M^−1^; (2) K_MIP,ZON_ = 16,849 M^−1^; CIP: (1) K_CIP,ZON_ = 137,492 M^−1^; (2) K_CIP,ZON_ = 30,213 M^−1^.

The same behaviour can be also attributed to the total number of binding sites for ZON where MIP values (N_MIP,ZON_ = 1.84 µmol g^−1^ (*r^2^* = 0.969); N_MIP,ZON_ = 16.88 µmol g^−1^ (*r^2^* = 0.900)) are higher than those of the CIP (N_CIP,ZON_ = 1.05 µmol g^−1^ (*r^2^* = 0.996); N_CIP,ZON_ = 3.77 µmol g^−1^ (*r^2^* = 0.881)).

### 2.4. Optimization of the MISPE Process for the Extraction of ZON

As indicated in Section 3.6, different parameters were evaluated for MISPE optimization: pH of the medium (P), mass of the nanoparticles (M), incubation time (I), volume of the elution solvent, and reusability of these particles (R).

#### 2.4.1. Incubation pH

ZON is a protonable compound in aqueous media that bears several phenolic groups that can be pH-dependent. The lower pKa for ZON is 7.4 and it corresponds to the hydroxyl group in the para position of the mycotoxin. This deprotonation is favoured for higher values of the pH. Then, the electrostatic interactions between the phenoxide with the primary amino groups of the MIP, generated after hydrolysis of the carbamate group which is protonated (pKa~10.3), can take place. In the evaluation of this effect, the chromatographic results revealed that an increase in pH can promote the interaction. Nevertheless, silica gel can be degraded in basic medium, which would negatively affect the selective retention of the mycotoxin. Figure 7A shows that the optimum value of the percentage of capture corresponding to the maximum retention of ZON (35%) in the MIP and minimum in the CIP (9%) was at pH = 7.5 using 50 mM phosphate as buffer solution.

#### 2.4.2. Mass of Fe_3_O_4_@SiO_2_@MIP Particles

A variation of the mass of the nanoparticles used for the solid phase extraction method was optimized, fixing a pH of 7.5 in the loading solution. Figure 7B shows that an increase in mass in turn increases the percentage capture of the analyte in both MIP (99%, RSD = 1%) and CIP (47%, RSD = 1%). This fact involves a soft decrease in the imprinting factor with respect to that calculated using the smallest mass, 1 mg. Nevertheless, prioritizing a higher value of the imprinting factor (2.96), the value of 1 mg of polymer with recoveries of 35% (RSD = 1%) in MIP and 12% (RSD = 1%) in CIP, was selected for further experiments.

#### 2.4.3. Incubation Time

Short incubation times are an important factor to optimize in a SPE method, in order to decrease the total time consumed in the analysis per sample. As Figure 7C shows, a 5 min incubation is enough to retain a relevant part of the ZON found in the loading solution. However, the equilibrium in MIP retention is obtained after 15 min of incubation with 35% (RSD = 2%) analyte capture. On the other hand, mycotoxin retention in CIP provides longer incubation times, which is indicative of slower kinetics from the non-specific bounds.

#### 2.4.4. Elution Solvent Volume

Due to the nature of the sorbent, ions of a different kind can be used to elute zearalenone. Tradicionally, for this type of interactions, a solvent able to disrupt the interactions between the functional monomer and the target molecules is used. In this case ethanol, due to the high solubility of the mycotoxin in this solvent, was selected with the addition of an organic modifier (trifluoroacetic acid), in order to break the electrostatic interactions due to the acidic media. To determine the optimal elution volume, consecutive additions of 2 mL of EtOH:TFA (98.5:1.5, *v*/*v*) were made on the nanoparticles already loaded with the analyte until the chromatographic peak corresponding to the analyte was no longer observed (Figure 7D). After 4 mL, the amount of analyte extracted was negligible.

#### 2.4.5. Cross Reactivity

To evaluate the selectivity of the imprinted polymer, structurally analogous molecules or mycotoxins that could be found in the same matrix as the target analyte were studied as possible interferents. Appendix A shows the high affinity of MIP for ZON (R = 100%). With respect to its similar structures, α-ZOL shows slightly lower recovery (R = 80%) while β-ZOL is significantly less retained (R = 40%) by MIP. In addition, a high cross-reactivity with AME (R = 80%) is observed due to its similar hydrophobicity with ZON, although compared to the non-imprinted material, it can be concluded that the retention of this analyte is not specific (R = 55%).

### 2.5. Sample Analysis

Compounds present in the matrix can severely affect the determination of the target analytes in the SPE procedures. To evaluate the appicability of the optimized SPE method to real samples, the optimized method was applied to the analysis of mineral, tap and river water samples. Due to the lack of certificate reference materials, water samples were fortified at three concentration levels of 12.5, 25 and 50 µg L^−1^. The samples were tested previously to confirm the absence of the analyte. The detection and the quantification limits (LOD, LOQ) obtained in these samples are summarized in Table 1. The limits of detection and quantification for ZON were calculated by using blank extracts at the lowest concentrations giving a signal-to-noise ratio of 3 and 10, respectively. As shown in Table 1, no significant differences were observed between the fortified and found concentration levels for all the samples tested.

Finally it is important to stress that the same material was reused more than 10 times without losing their properties.

## 3. Materials and Methods

### 3.1. Chemicals

Zearalenone was purchased from abcr (Karlsruhe, Germany). Trifluoroacetic acid (TFA) was obtained from Fluorochem (Hadfield, UK). HPLC water was purified through a Milli-Q system from Millipore (Bedford, MA, USA). HPLC-grade dimethyl sulfoxide (DMSO), absolute ethanol, ammonia 28% and 1,4-dioxane were acquired from VWR (Paris, France). HPLC-grade methanol ≥99.8% was supplied by Fisher Chemical (Merelbeke, Belgium). Disodium phosphate Na_2_HPO_4_, sodium phosphate monobasic NaH_2_PO_4_ and HPLC-grade acetonitrile (ACN) were from Honeywell (Langenhagen, Germany). Iron (III) chloride hexahydrate and n-hexane were obtained from Analyticals Carlo Erba (Barcelona, Spain). Extra pure ethylene glycol, hydrochloric acid (36.5–38%), ethyl acetate (AcEt), triethylamine (Et_3_N) and silica gel were supplied by Scharlab (Barcelona, Spain). Polyethylene glycol (PEG), hexadecyltrimethylammonium bromide (CTAB) was from Fluka Chemicals (Feltham, United Kingdom). Oleic acid, tetraethyl orthosilicate (TEOS) (≥99.0%), toluene HPLC grade and sodium bicarbonate (NaHCO_3_) were obtained from Sigma Aldrich (Madrid, Spain). 1,1′-carbonyldiimidazole, 2,4-dihydroxybenzoic acid, cyclododecanol, *N*,*N*-dimethylformamide (DMF) anhydrous, 1,8-diazabicyclo [5.4.0]undec-7-ene (DBU), chloroform (≥99.0%) (CHCl_3_), 3-(triethoxysilyl)propyl isocyanate and methyl 4-dihydroxybenzoate were from Acros Organics (Geel, Belgium). Dichloromethane (CH_2_Cl_2_) and sodium sulfate (Na_2_SO_4_) were purchased from Fischer Scientifics (Fair Lawn, NJ, USA). Argon gas was supplied by Carburos metálicos (Barcelona, Spain).

### 3.2. Apparatus

Microwave (MW) reactor 850-W Anton Paar 200 (Graz, Austria) and its own MW vials were used for the nanoparticle synthesis. In order to disperse homogeneously the particles into solution an Ultrasonicator VCX 130 PB Vibracell (Newtown, CT, USA) was used. A Fisherbrand Classic vortex mixer was also used during this work. Silica polymerization requires homogeneous heat that was provided by an Heraeus Oven by Thermo Scientific (Whaltham, MA, USA). All the compounds were carefully weighed using a Mettler Toledo analitycal balance (Barcelona, Spain). The pH of the buffer solutions was adjusted with a Crison GLP22+ pH meter (Barcelona, Spain). All the buffers were prepared by dissolving the corresponding volume of the different phosphate salt solutions (50 mM) until reaching desired pH values. These solutions were filtered through a 0.45 µm nylon membrane before using. The chemical structure of the compounds synthetized was firstly studied by ^1^H-NMR and ^13^C-NMR (Bruker Avance DPX300 MHz, UCM NMR Central Instrumentation Facilities) at room temperature using CDCl_3_ as solvent. Finally, the structure was confirmed by using mass spectroscopy which was performed by using ion trap Bruker equipment (HCT ultra PTM Discovery, UCM Central Instrumentation Facilities, Madrid, Spain) setting ESI as ion source type, starting the scanning at 200 *m*/*z* and finishing at 1000 *m*/*z*. Fourier transform infrared (FTIR) spectra were recorded using the compact FTIR Bruker ALPHA (Madrid, Spain), setting the scanning range from 4000 cm^−1^ to 650 cm^−1^. In addition, every step during the nanoparticle synthesis was characterized using a transmission electron microscope (JEM-1400 PLUS, JOEL, Tokyo, Japan) coupled with a SDD microanalysis system in order to measure the size, structure and composition. The crystal structure of the different products obtained was defined by X-ray Powder Diffraction (XRD) using Philips X’pert MPD equipment (Philips, Amsterdam, The Netherlands) adopting Cu Kα radiation (1.5405 Å). Thermogravimetric studies (TGA) were performed with a PerkinElmer apparatus (STA 6000, PerkinElmer, Waltham, MA, USA) putting the samples into a platinum crucible and heated with an increase of 10 °C per min from 50 to 800 °C in a 20 cm^3^ min^−1^ N_2_ flow.

### 3.3. HPLC Analysis

High performance liquid chromatography equipment combined with multiwave detector and fluorescent detector (HPLC-FLD) was used to perform both qualitative and quantitative measurements. All chromatographic measurements were performed, using a quaternary pump Agilent 1100 high performance liquid chromatograph (Santa Clara, CA, USA) using a C18 ZORBAX column (150 mm × 4.6 mm, 3–5 µm) by Agilent as stationary phase. The mobile phase used for chromatographic analysis consisted of acetonitrile containing 0.1% (*v*/*v*) TFA, as solvent A, and Milli-Q water containing 0.1% (*v*/*v*) TFA as solvent B. The analysis was performed in 50:50 isocratic mode (A:B) at a flow rate of 1 mL min^−1^. All compounds were detected at λ_exc_ = 270 nm, λ_em_ = 452 nm and the column temperature was set at 20 °C. The injection volume was 10 µL. The ZON eluted at around 7 min.

### 3.4. Silane Derivate Synthesis

#### 3.4.1. Synthesis of Cyclododecyl 2,4-dihydroxybenzoate (CDHB)

This molecule was synthesized as described previously by Urraca et al. [45].

#### 3.4.2. Synthesis of Cyclododecyl 2-hydroxy-4-(3-triethoxysilylpropylcarbamoyloxy) Benzoate (Surrogate-MIP)

In this work, the surrogate used previously has been modified and bound to a silane in order to incorporate it into the polymer matrix. In a double necked flask was placed, under stirring, a mixture of cyclododecyl 2,4-dihydroxybenzoate (424 mg; 1.3 mmol) dissolved in anhydrous CHCl_3_ (15 mL) and dropwise added triethylamine (200 mg; 1.8 mmol). After 5 min flowing argon, 3-(triethoxysilyl)propyl isocyanate (247 mg; 0.9 mmol) was added and the reaction was kept under reflux conditions for 48 h. Then, the reaction product was washed with H_2_O (3 × 15 mL), dried with Na_2_SO_4_ and evaporated under reduced pressure. Further purification by flash chromatography in silica gel with n-hexane and ethyl acetate (18:1, *v*/*v*) yielded the desired product (176.6 mg, 35.1% yield). IR ν(cm^−1^): 3336, 2936, 1715, 1685, 1535, 1494, 1267, 1243, 1168, 1081, 959 and 772. NMR chemical shifts (δ) are expressed in parts per million relative to internal tetramethylsilane. Abbreviations for peak patterns: s(singlet), d (doublet), dd (double doublet), t (triplet), m (multi-plet), bs (broad singlet):^1^H-NMR (CDCl_3_, 300 MHz): δ 0.67 (t, 2H, J = 8.01 Hz); 1.17 (t, 9H, J = 6.96 Hz) 1.30 (bs, 20H); 1.59 (bs, 2H); 3.21 (q, 2H, J = 6.58 Hz); 3.77 (q, 6H, J = 7.00 Hz); 5.21 (qd, 1H, J = 7.16; 4.62 Hz); 5.36 (t, 1H, J = 5.92 Hz); 6.62 (dd, 1H, J = 8.72, 2.30 Hz); 6.68 (d, 1H, 2.30 Hz); 7.76 (d, 1H, J = 8.72 Hz); 10.97 (s, 1H). ^13^C-NMR (CDCl_3_, 75 MHz):δ 7.9; 18.3; 21.0; 23.5; 24.3; 24.4; 24.5; 29.4; 32.6; 43.4; 57.7; 73.7; 110.1; 110.3; 112.9; 130.7; 153.5; 156.4; 162.9; 169.7. Mass spectrum (*m*/*z*, %): 651 (8), 567 (8), 566 (18) [M^−^], 355 (6), 321 (8), 320 (47), 319 (100).

#### 3.4.3. Synthesis of Methyl 4-(3-triethoxysilylpropylcarbamoyloxy) Benzoate (Surrogate-CIP)

To a solution of methyl 4-hydroxybenzoate (148.4 mg, 0.975 mmol) in anhydrous CHCl_3_ (15 mL), triethylamine (200 mg; 1.3 mmol) was added. After 5 min flowing argon, 3-(triethoxysilyl)propyl isocyanate (160 mg; 0.650 mmol) was added and the reaction was kept under reflux conditions for 24 h. Then, the reaction product was washed with H_2_O (3 × 15 mL), dried with Na_2_SO_4_, and evaporated under reduced pressure. Further purification by flash chromatography in silica gel with n-hexane and ethyl acetate (20:1, *v*/*v*) yielded the desired product (169.3 mg, 65.4% yield).^1^H-NMR (CDCl_3_, 300 MHz): δ 0.69 (t, 2H, J = 7.98 Hz); 1.24 (t, 9H, J = 7.00 Hz); 1.72 (dq, 2H, 8.25; 6.96 Hz); 3.28 (q, 2H, J = 6.58 Hz); 3.84 (q, 6H, J = 7.00 Hz); 3.90 (s, 3H); 5.47 (bs, 1H); 7.20 (d, 2H, J = 8.76 Hz); 8.04 (d, 2H, 8.74 Hz).

### 3.5. MIPs/CIPs Fabrication

#### 3.5.1. Fe_3_O_4_ and Fe_3_O_4_@SiO_2_ Magnetic Nanoparticles

Fe_3_O_4_ magnetic bare particles were synthesized by a described solvothermal method with slight modifications [39]. Briefly, an amount of 0.675 g of FeCl_3_·6H_2_O was mixed with 1.8 g of sodium acetate in 20 mL of ethyleneglycol. Once the mixture was homogeneous, 0.45 mL of polyethyleneglycol was added. Next, the reaction was shaken at room temperature for 30 min and transferred to a 30 mL microwave vial. The conditions fixed at the MW system were: 200 °C; 8 min; 600 rpm. The resulting black nanoparticles (NPs) were washed twice with 250 mL of methanol and twice with 250 mL of water, removed from the solution using a permanent magnet, and dried under vacuum at 60 °C.

In order to cover the Fe_3_O_4_ magnetic NPs previously obtained with a homogeneous silica layer, the traditional Stöber method was slightly adapted [48,49]. First, 20 mg of Fe_3_O_4_ nanoparticles were put into a vial where 300 µL of oleic acid was added, a necessary step for avoiding high aggregation rates. After 15 min of sonication, 500 µL of EtOH was carefully poured into the vial, and 30 min of rolling inside a 60 °C oven was then required. This particle suspension was completely dispersed into 10 mL of EtOH using ultrasound equipment again. This procedure has to be repeated dispersing our mixture into 20 mL of a 6 mM CTAB solution for 30 min (NPs solution); by adding the surfactant agent, the excessive aggregation of the Fe_3_O_4_ nanoparticles is avoided. Meanwhile, a mixture of 70 mL of EtOH and 175 mL of 6 mM CTAB solution was placed in a 250 mL borosilicate bottle and vigorously stirred. The Stöber method requires a basic medium for the silica to polymerize; hence, once the solution was completely homogeneous, 0.2 mL of 28% pure ammonia was added. The NPs mixture is poured into the borosilicate bottle containing the basic solution, and perfectly dispersed, again using ultrasound sonication. Finally, 250 µL of TEOS were added dropwise and the final solution was maintained rolling in the oven at 60 °C for 72 h. The brownish particles obtained were washed, and a permanent magnet was used in order to remove possible impurities: 2 × 20 mL of pure EtOH; 2 × 10 mL of 0.1 M HCl in EtOH solution, in order to eliminate CTAB from the pores and finally 2 × 20 mL of pure EtOH again. The particles were dried under vacuum at 60 °C.

#### 3.5.2. Preparation of the Functionalized Magnetic Nanoparticles (Fe_3_O_4_@SiO_2_@MIP/CIP)

To graft the previously synthesized modified silica molecule (MIP/CIP) on the Fe_3_O_4_@SiO_2_ surface, it was first necessary to disperse 30 mg of the particles into approximately 5 mL of dry toluene via sonication during 15 min. Next, the MIP/CIP-surrogate was added (1 mmol) and the mixture was refluxed during 24 h under argon atmosphere. The obtained Fe_3_O_4_@SiO_2_@MIP/CIP was separated using an external magnetic field and washed with MeOH (3 × 15 mL).

#### 3.5.3. Surrogate Extraction

Since the covalent bounds formed by the reaction between the isocyanate and phenolic groups are stable at room temperature, but labile at high temperatures [50,51], 30 mg of the particles Fe_3_O_4_@SiO_2_@MIP/CIP suspended on a 1,4-dioxane/H_2_O (7:1; *v*/*v*) mixture were refluxed during 24 h to remove the template from the structured silica polymer. During this reaction, the urethane bonds were dissociated resulting in the isocyanate groups, which are sensitive to the water present in the solvent mixture ending up in the final amino groups. These particles were washed with MeOH (3 × 15 mL). This procedure was repeated several times until no template molecule was detected by HPLC.

### 3.6. Evaluation of Parameters for MISPE Process

Parameters directly dependent on the selective retention capacity of the mycotoxin in Fe_3_O_4_@SiO_2_@MIP were evaluated. The parameters studied were: (a) pH of the medium (P); (b) mass of the nanoparticles (M); (c) incubation time (I); (d) volume of the elution solvent (E), and (e) reusability of these particles (R). The experimental domain used to evaluate the influence of the experimental variables was: (a) P: 7–8.5 in 0.050 mol L^−1^ phosphate buffer; (b) M: 1–12 mg; (c) I: 5–240 min; (d) E: 2–4 mL of EtOH-TFA, 98.5:1.5 *v*/*v*, and (e) R: 2–10 times. In all cases, the sample loading volume and ZON concentration were kept constant at 1 mL and 150 µg L^−1^, respectively. Simultaneously, the response of the control polymer (CIP) to these same parameters was studied to evaluate non-specific interactions.

### 3.7. MIPs and CIPs Rebinding Performance

Into 2 mL HPLC vials, 1 mL of the correspondent zearalenone buffered solution and the required amount of functionalized NPs were added. Both, the analyte and the adsorbent quantity would depend on the experiment being performed in each case. Then, the mixture was sonicated for 30 s and shaked for a fixed period of time. After the incubation step, the solution was poured apart from the nanoparticles by magnetic separation and filtered using a 0.45 µm filter. The solution was then analysed by HPLC-FLD, using the analytical parameters previously described. The percentage of ZON adsorbed into the surface of the nanoparticles was calculated by using the following equation:(1)$$Adsorbed\;\left(\%\right)=\frac{Ci-Ce\;}{Ci}\;\text{×}\;100$$
where *Ci* and *Ce* correspond to the initial and the final concentration (mg L^−1^) of ZON present in the buffered solution, respectively. Furthermore, the amount of adsorption at equilibrium, *qe* (mg g^−1^) was also calculated, using the following formula:(2)$$qe=\frac{Ci-Ce\;}{m}\;\times \;V$$

The buffered solution volume is referred as V (mL) while m is the nanoparticle mass used as adsorbent (g).

### 3.8. Real Sample Preparation

In this work, three different kind of samples were analysed. The first sample was a commercially obtained natural mineral water from Fuentevera (Toledo, Spain). The second sample was a tap water sample from the Community of Madrid (Madrid, Spain) and the last one was a water sample from the river Cerezuelo Cazorla (Jaén, Spain). The water samples were centrifuged at 7000 rpm for 10 min. The supernatant was brought to pH 7.5 with 50 mM phosphate buffer. The extracts were spiked at three concentration levels of ZON (12.5, 25 and 50 µg L^−1^). All experiments were performed in triplicate.

## 4. Conclusions

As we have shown in this work, the polymerisation procedure carried out on the synthetic magnetic cores allowed us to obtain totally homogeneous, spherical and individual polymeric nanoparticles without the formation of any type of aggregate between them during polymerisation. The procedure allows the total elimination of any remaining polymer that does not belong to the polymeric layer formed around the nanoparticles and thus avoids possible non-specific interactions on the part of the nanoparticles. It also provides a correct and total elution of the mycotoxin retained by the nanoparticles during the tests, as the elution solvent can better access the entire polymer to extract the analyte completely. These polymeric nanoparticles with magnetic properties have been successfully applied to the analysis of real water samples of different origins and characteristics, obtaining optimal results with a high degree of precision, accuracy and repeatability. Finally, the results obtained in this paper demonstrate that the zearalenone analogue molecule used as a template molecule for the synthesis of the polymer is totally suitable as surrogate of ZON for obtaining MIPs. Finally, Fe_3_O_4_@SiO_2_@MIP particles were able to recognise not only zearalenone but also other mycotoxins of the zearalenone family such as α- and β-zearalenol.

## Figures and Tables

**Figure 1 molecules-27-08166-f001:**
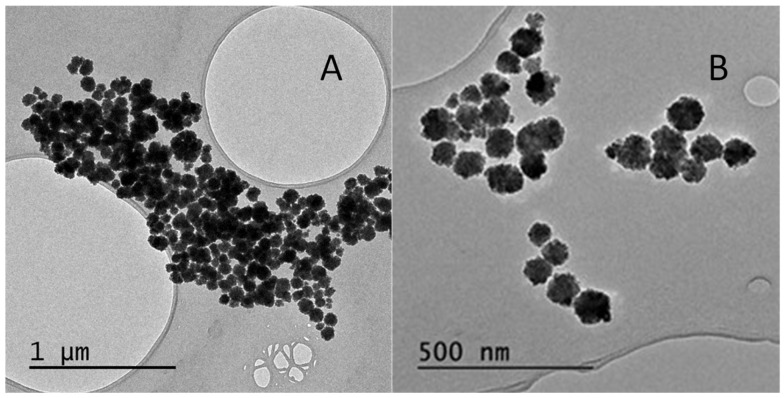
Transmission electron microscopy micrographs of the synthesis methods used for Fe_3_O_4_ nanoparticles: (**A**) Method 1, solvothermal method; (**B**) Method 2, microwave reactor.

**Figure 2 molecules-27-08166-f002:**
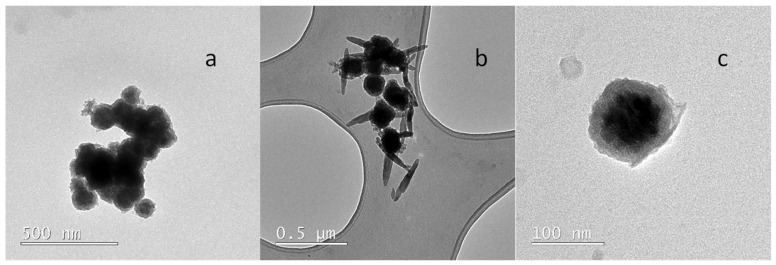
Transmission electron microscopy micrographs of the synthesis methods used on Fe_3_O_4_@SiO_2_: (**a**) Method A; (**b**) Method B; (**c**) Method C.

**Figure 3 molecules-27-08166-f003:**
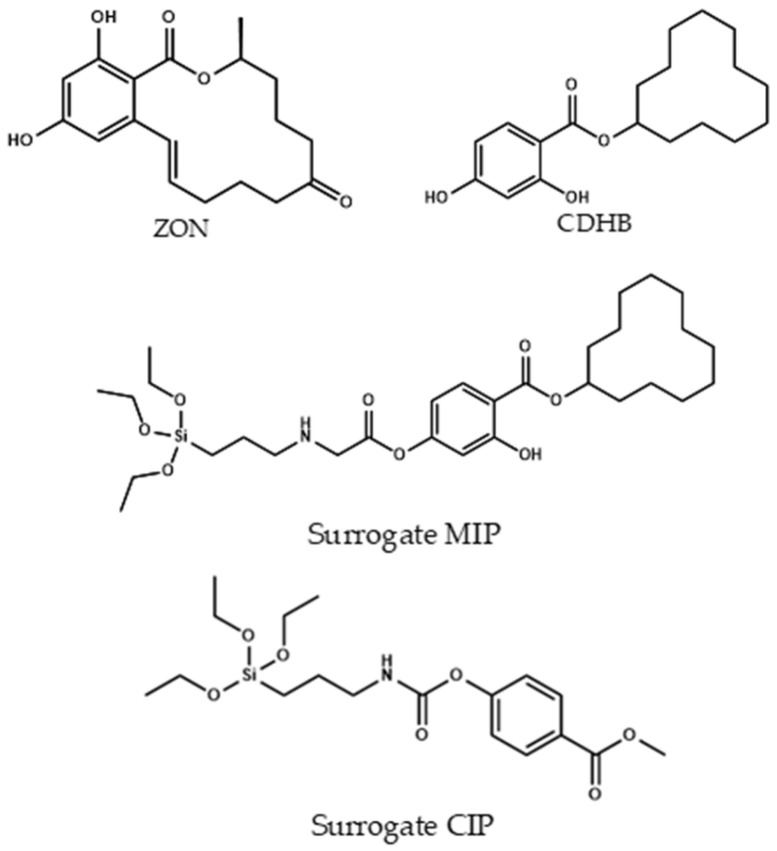
Chemical structures of zearalenone (ZON), CDHB, and surrogate-MIP and surrogate-CIP.

**Figure 4 molecules-27-08166-f004:**
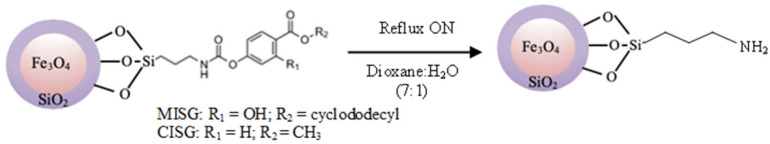
Reaction scheme for the elimination of the template molecule.

**Figure 5 molecules-27-08166-f005:**
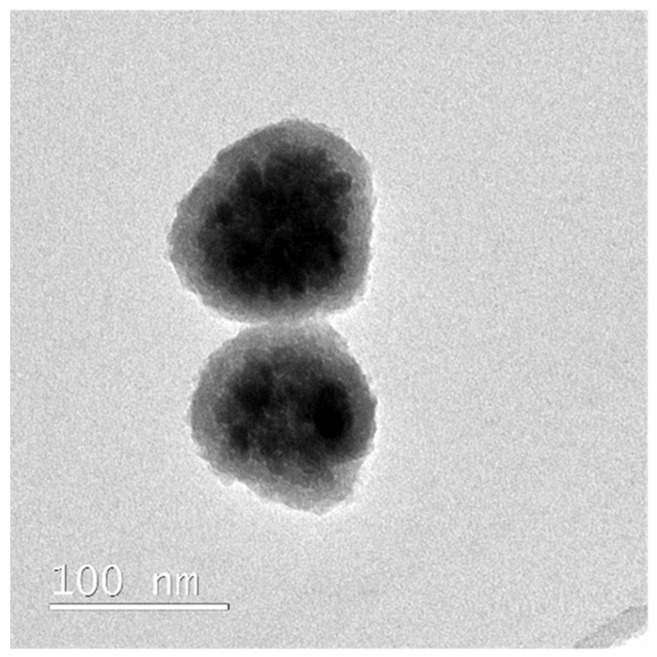
Transmission electron microscopy micrograph of the synthesis methods used on Fe_3_O_4_@SiO_2_@MIP.

**Figure 6 molecules-27-08166-f006:**
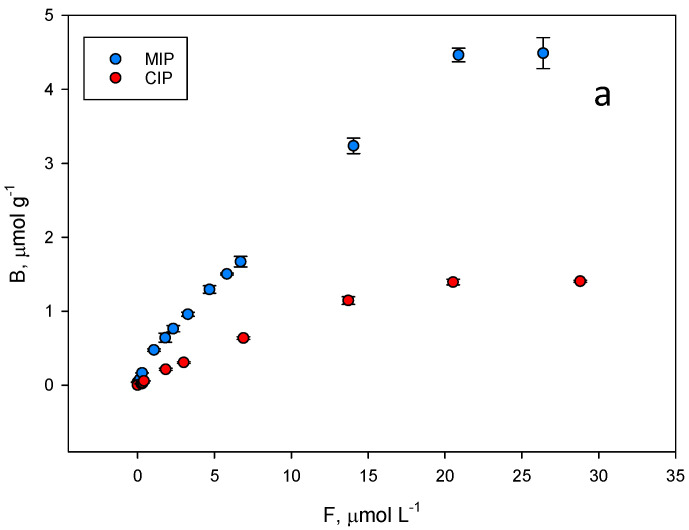
(**a**) Equilibrium binding isotherms for ZON to the MIP (blue) and to the corresponding CIP (red) in phosphate buffer (50 mM, pH 7.5). (**b**) Scatchard fitting for the MIP and (**c**) the CIP. High affinity, blue lines. Low affinity, red lines.

**Figure 7 molecules-27-08166-f007:**
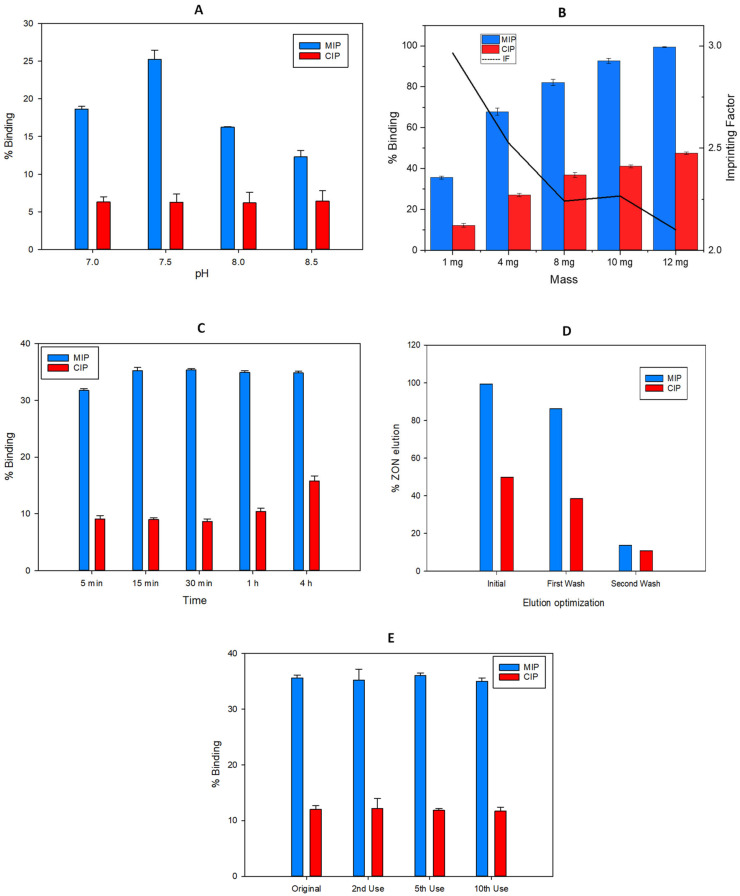
Optimization in terms of binding recovery (%) of the MISPE parameters: (**A**) pH incubation, (**B**) Mass of Fe_3_O_4_@SiO_2_@MIP particles, (**C**) Incubation time, (**D**) Elution solvent volume and (**E**) Material reuse, for the extraction of ZON.

**Table 1 molecules-27-08166-t001:** LODs, LOQs, mean concentration levels and confidence levels (95%) for the analysis of ZON in water samples spiked at three different levels (*n* = 3).

Spiked Conc. (µg kg^−1^)	R_A_ in Mineral Water (µg kg^−1^)	R_A_ in Tap Water (µg kg^−1^)	R_A_ in River Water (µg kg^−1^)
12.5	12 ± 1	11 ± 2	10 ± 3
25	16 ± 11	17 ± 9	14 ± 9
50	32 ± 12	30 ± 4	31 ± 4
LOD	2.3	2.4	2.4
LOQ	7.7	8.0	8.1

## Data Availability

The data presented in this study are available on request from the corresponding authors.

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
