# Peer review of "Magnetic Core-Shell Nanoparticles Using Molecularly Imprinted Polymers for Zearalenone Determination"

_molecules, 2022, doi:10.3390/molecules27238166_

Round 1
Reviewer 1 Report
I think that the study merits publication because includes a very detailed description of the synthesis and characterization of a novel magnetic composite material selective to zearalenone (ZEN) mycotoxin and that can be applied to solid phase extraction for the determination of the compound by HPLC-FLD.
A revision is recommended to improve the clarity of the results described in the manuscript. Here you can find some of my comments:
1) Please, revise the nomenclature and try to use the same terminology throughout the manuscript and figure captions to facilitate reading. E.g. Fe3O4, Fe3O4@SiO2 and Fe3O4@SiO2@MISG.
Another example: In Figure S2 use “Fe3O4@SiO2@MISG” instead of “Fe3O4@SiO2 coated with sol-gel MIP”
2) The use of the word MIP in the text is confusing sometimes. Please, rewrite lines 13-16 in the abstract “The MIP….molecule”. I suggest writing: “ The nanobeads were prepared….. with a second layer (MIP) using cyclododecil…This compound was used…”
In 2.4.2 and 2.4.3 the authors use the words MIP and CIP when they are explaining the synthesis of the functional monomers FMSG-M and FMSG-C (see Figure 3). But section 2.5 is entitled "MIPs/CIPs fabrication" again.
3) Mention also in the text the definition of the acronyms or abbreviations you are using, e.g. “2.4.1 Synthesis of cyclododecyl 2,4-dihydroxybenzoate (CDHB)” , MISG, MISGs (Fe3O4@SiO2@MISG nanobeads), CISGs, etc…
4) Revise figure captions and their mentions in the text
5) Figure 1. There is a mistake in this figure; (a) and (b) are identical
6) Figure 6. Figure 6 about the optimization of the solid phase extraction procedure described in the text (section 3.4) is not present in the pdf file.
7) Please, revise the manuscript to correct some grammar mistakes and typos, e.g.: rapprochement (line 80), shacked (line 239), ultrasound sonication (line 246), CHDB (line 352), taked (line 358), be also be (line 462), should can (line 477), etc. Table S1. Replace commas by decimal dots
Author Response
Thank you very much for your comments and we agree with many of them. The text has been thoroughly revised and corrected. We also apologize for the large number of minor errors.The writing style has also been corrected.
1) Please, revise the nomenclature and try to use the same terminology throughout the manuscript and figure captions to facilitate reading. E.g. Fe3O4, Fe3O4@SiO2 and Fe3O4@SiO2@MISG.
Another example: In Figure S2 use “Fe3O4@SiO2@MISG” instead of “Fe3O4@SiO2 coated with sol-gel MIP”.
Corrected. Many of this terms have been unified in text.
2) The use of the word MIP in the text is confusing sometimes. Please, rewrite lines 13-16 in the abstract “The MIP….molecule”. I suggest writing: “ The nanobeads were prepared….. with a second layer (MIP) using cyclododecil…This compound was used…”
Corrected in the text.
In 2.4.2 and 2.4.3 the authors use the words MIP and CIP when they are explaining the synthesis of the functional monomers FMSG-M and FMSG-C (see Figure 3). But section 2.5 is entitled "MIPs/CIPs fabrication" again.
Corrected in the text. Section 2.4.2 and 2.4.3 correspond to the synthesis of the surrogates molecules. It has been included in the text.
3) Mention also in the text the definition of the acronyms or abbreviations you are using, e.g. “2.4.1 Synthesis of cyclododecyl 2,4-dihydroxybenzoate (CDHB)” , MISG, MISGs (Fe3O4@SiO2@MISG nanobeads), CISGs, etc…
Corrected in the text. All the acronyms have been explained.
4) Revise figure captions and their mentions in the text
Corrected in the text
5) Figure 1. There is a mistake in this figure; (a) and (b) are identical
Figure 1b has been changed.
6) Figure 6. Figure 6 about the optimization of the solid phase extraction procedure described in the text (section 3.4) is not present in the pdf file.
Figures and tables were supported in a different file. Maybe that is the reason. Now it has been corrected.
7) Please, revise the manuscript to correct some grammar mistakes and typos, e.g.: rapprochement (line 80), shacked (line 239), ultrasound sonication (line 246), CHDB (line 352), taked (line 358), be also be (line 462), should can (line 477), etc. Table S1. Replace commas by decimal dots.
All these minor errors have been corrected.
Reviewer 2 Report
This manuscript investigated the synthesis of molecularly imprinted magnetic nano-beads for the selective extraction (MISPE) of zearalenone mycotoxin in river and tap waters and further analysis by high-performance liquid chromatography (HPLC) with fluorescence (FLD) detection. The content of the manuscript was relatively sufficient, but there were still some deficiencies in innovation and there are some grammatical errors in the article. Following problems need to be paid attention.
1. Please supplement the graphic abstract of the synthetic route.
2. Line 96, change MMPIs to MMIPs.
3. Line 146, 0.45 m nylon membrane?
4.
5. Line 157-158, XDR should be changed to XRD.
6. Line 218, what is CIP? The author should specify it in advance. All abbreviations in the text should be explained in advance.
7. The chart used in this paper is too simple. Could you use a more advanced chart to explain the data?
8. In 2.5.1, Is Fe3O4 synthesized successfully? The author should add XRD to make the corresponding detection and add VSM detection to explain the magnetic properties of the material.
9. Line 267, what is the 1 mmolg-1?
10. Line 327, figure S1a?
11. The diameter of the biretta in Figure 1c is 200 nm, but the particle size is about 20 nm?
12. Line 345, the particle size in Figure 2C is far from 20nm. Please review it.
13. The FTIR image needs to be optimized to mark each peak.
14. Line 380-381, “figure 2 and Figure 5”, this happens many times in the text, and the case of the letters should be uniform.
15. All illustrations should be modified to be more beautiful and the font should be uniform.
16. In 3.2.3, Figure S5 should be Figure S2, please check it.
17. In 3.2.4, the content of washed carbon was still higher than the content of Fe3O4@SiO2, how to assure that the template was washed cleanly.
18. In 3.3, Figure 5 should be Figure S5, please check it.
19. Figure 6 is not found in the article, please check it.
20. The reference formats were confusing.
Author Response
Thank you very much for your comments and we agree on many of them. The text has been thoroughly revised and corrected. We also apologize for the large number of minor errors.The writing style has also been corrected.
This manuscript investigated the synthesis of molecularly imprinted magnetic nano-beads for the selective extraction (MISPE) of zearalenone mycotoxin in river and tap waters and further analysis by high-performance liquid chromatography (HPLC) with fluorescence (FLD) detection. The content of the manuscript was relatively sufficient, but there were still some deficiencies in innovation and there are some grammatical errors in the article. Following problems need to be paid attention.
- Please supplement the graphic abstract of the synthetic route.
Graphical abstract has been modified.
- Line 96, change MMPIs to MMIPs.
Corrected
- Line 146, 0.45 m nylon membrane?
Corrected
- Line 157-158, XDR should be changed to XRD.
Corrected
- Line 218, what is CIP? The author should specify it in advance. All abbreviations in the text should be explained in advance.
Many abreviatons have been deleted. All abreviations are explained now.
- The chart used in this paper is too simple. Could you use a more advanced chart to explain the data?
Corrected
- In 2.5.1, Is Fe3O4synthesized successfully? The author should add XRD to make the corresponding detection and add VSM detection to explain the magnetic properties of the material.
It is. Both graphs are in the sup. Material.
- Line 267, what is the 1 mmolg-1?
Corrected
- Line 327, figure S1a?
Corrected
- The diameter of the biretta in Figure 1c is 200 nm, but the particle size is about 20 nm?
It has been clarified in the text. 20 nm is the shell size. Figure 1c does not exist.
- Line 345, the particle size in Figure 2C is far from 20nm. Please review it.
It has been clarified in the text. 20 nm is the shell size. Figure 2c has been replaced
- The FTIR image needs to be optimized to mark each peak.
Corrected
- Line 380-381, “figure 2 and Figure 5”, this happens many times in the text, and the case of the letters should be uniform.
Corrected
All illustrations should be modified to be more beautiful and the font should be uniform.
Corrected
- In 3.2.3, Figure S5 should be Figure S2, please check it.
Corrected
- In 3.2.4, the content of washed carbon was still higher than the content of Fe3O4@SiO2,how to assure that the template was washed cleanly.
After hydrolysis, the molecule that acts as functional monomer still bears carbons. This small percentage of Carbons remnant is normal.
- In 3.3, Figure 5 should be Figure S5, please check it.
Corrected
Figure 6 is not found in the article, please check it.
Corrected
- The reference formats were confusing.
Corrected
Reviewer 3 Report
Journal: Molecules
Manuscript ID: molecules-2008144
Title: Magnetic Core-Shell Nanoparticles Using Molecularly Imprinted Polymers for Zearalenone Determination
In this work, the authors reported magnetic core-shell nanoparticles based on molecularly imprinted polymers (MIPs) for detecting zearalenone. Various characteristics such as morphology, crystallinity, chemical component, thermogravimetric analysis, and adsorption property were abundantly investigated in the manuscript. This work is interesting. However, there are many errors throughout the manuscript. The current version needs a lot of improvements.
1. Figure 1 a,b images are the same. In addition, the authors noted with respect to in Figure 1.
“In addition, the authors noted with respect to in Figure 1. were obtained with an average diameter similar to that obtained previously 85 ± 11 nm.”
As shown In Figure 2c, why did the size of the core increase when fabricating the shell of 20 ± 4 nm using Method C?
2. In Figures 3 and 4, the resolution of the chemical structures is very different. It must be modified appropriately.
3. In general, MIPs consist of template, functional monomer and cross-linker. For high stability of the cavities in the polymer matrix, more than 80% of the crosslinking agent is used. In this work, what is the composition and chemical structure of the MIP matrix in Fe3O4@SiO2@MIP/CIP?
4. P. 10, L. 389 XRD patterns are presented in Figure S2. It is also difficult to identify the data. For a better understanding, the figure should be modified [10.1016/j.jhazmat.2014.05.013].
5. Not Figure 5 (P. 11 L. 450 and 455), the Scatchard plot is shown in Figure S5. In addition, since binding isotherm is important information in this study, it should be presented in the manuscript instead of supporting information. In addition, different models (e.g., Freundlich model) can be applied to compare the fit.
6. Where is Figures 6a-d?
7. The adsorption characteristics should be compared with other reported literatures.
8. There are a lot of minor errors.
P. 5, L. 203, 3-(Triethoxysilyl)propyl isocyanate -> 3-(triethoxysilyl)propyl isocyanate
P. 5, L. 218, Synthesis of Methyl 4-(3-triethoxysilylpropylcarbamoyloxy) benzoate (CIP) -> Synthesis of methyl 4-(3-triethoxysilylpropylcarbamoyloxy) benzoate (CIP)
P. 5, L. 219, methyl 4- hydroxybenzoate -> methyl 4-hydroxybenzoate
P. 5, L. 221, 3-(Triethoxysilyl)propyl isocyanate -> 3-(triethoxysilyl)propyl isocyanate
P. 5, L. 247, 60°C -> 60 °C
P. 5, L. 250, 6mM -> 6 mM
P. 6, L. 258, 60°C for 72h -> 60 °C for 72 h
P. 6, L. 259, 2x20mL
P. 6, L. 277, 24h
P. 6, L. 280, 15mL
P. 7, L. 320, 190 ⁰C
P. 7, L. 327, figure S1a -> Figure 1a
P. 7, L. 330, 200 ⁰C
P. 7, L. 332, figure 1
P. 8, L. 346, figure 2c
P. 8, L. 354, figure 3
P. 8, L. 357, figure 4
P. 8, L. 362, Methyl 4-(3-triethoxysilylpropylcarbamoyloxy)
P. 10, L. 420 and 421, ⁰C
P. 11, L. 439, figure S4
Author Response
Thank you very much for your comments and we agree on many of them. The text has been thoroughly revised and corrected. We also apologize for the large number of minor errors.The writing style has also been corrected.
In this work, the authors reported magnetic core-shell nanoparticles based on molecularly imprinted polymers (MIPs) for detecting zearalenone. Various characteristics such as morphology, crystallinity, chemical component, thermogravimetric analysis, and adsorption property were abundantly investigated in the manuscript. This work is interesting. However, there are many errors throughout the manuscript. The current version needs a lot of improvements.
- Figure 1 a,b images are the same. In addition, the authors noted with respect to in Figure 1.
“In addition, the authors noted with respect to in Figure 1. were obtained with an average diameter similar to that obtained previously 85 ± 11 nm.”
As shown In Figure 2c, why did the size of the core increase when fabricating the shell of 20 ± 4 nm using Method C?
Figures have been changed to clarify these points.
- In Figures 3 and 4, the resolution of the chemical structures is very different. It must be modified appropriately.
Corrected
- In general, MIPs consist of template, functional monomer and cross-linker. For high stability of the cavities in the polymer matrix, more than 80% of the crosslinking agent is used. In this work, what is the composition and chemical structure of the MIP matrix in Fe3O4@SiO2@MIP/CIP?
It is right. But this MIP is not a conventional MIP. The silane (surrogate molecule) acts as template and functional monomer, thus the stoichiometry is 1:1. But also it can be considered a cross-linker due to through the silanol groups is bound to the TEOS layer. Even then, TEOS could be a part of the cross-linker. Thereby a ratio, template: functional monomer: cross-linker can be not estimated.
- P. 10, L. 389 XRD patternsare presented in Figure S2. It is also difficult to identify the data. For a better understanding, the figure should be modified [10.1016/j.jhazmat.2014.05.013].
It has been modified.
- Not Figure 5 (P. 11 L. 450 and 455), the Scatchard plot is shown in Figure S5. In addition, since binding isotherm is important information in this study, it should be presented in the manuscript instead of supporting information. In addition, different models (e.g., Freundlich model) can be applied to compare the fit.
They have been included in the text.
- Where is Figures 6a-d?
Corrected.
- The adsorption characteristics should be compared with other reported literatures.
We agree in part with the referee. The adsorption characteristics for other papers (even for papers of any of these authors) can be calculated in a different way and It can be “dangerous” to make this kind of comparisons. Moreover it is the first time that a MIP is reported to determine ZON in aqueous media, thus the comparison can be a little bit “tricky”.
- There are a lot of minor errors.
- 5, L. 203, 3-(Triethoxysilyl)propyl isocyanate -> 3-(triethoxysilyl)propyl isocyanate
- 5, L. 218, Synthesis of Methyl 4-(3-triethoxysilylpropylcarbamoyloxy) benzoate (CIP) -> Synthesis of methyl 4-(3-triethoxysilylpropylcarbamoyloxy) benzoate (CIP)
- 5, L. 219, methyl 4- hydroxybenzoate -> methyl 4-hydroxybenzoate
- 5, L. 221, 3-(Triethoxysilyl)propyl isocyanate -> 3-(triethoxysilyl)propyl isocyanate
- 5, L. 247, 60°C -> 60 °C
- 5, L. 250, 6mM -> 6 mM
- 6, L. 258, 60°C for 72h -> 60 °C for 72 h
- 6, L. 259, 2x20mL
- 6, L. 277, 24h
- 6, L. 280, 15mL
- 7, L. 320, 190 ⁰C
- 7, L. 327, figure S1a -> Figure 1a
- 7, L. 330, 200 ⁰C
- 7, L.332, figure 1
- 8, L. 346, figure 2c
- 8, L. 354, figure 3
- 8, L. 357, figure 4
- 8, L.362, Methyl 4-(3-triethoxysilylpropylcarbamoyloxy)
- 10, L. 420 and 421, ⁰C
- 11, L. 439, figure S4
Thank you so much for these minor errors. All them have been corrected.
Reviewer 4 Report
In this manuscript preparation and characterization process of magnetic core-shell molecularly imprinted nanoparticles for zearalenone determination was described. Obtained material was tested as solid phase sorbent during determination of analyte in water samples. The topic of the manuscript is interesting, especially in connection with interesting synthetic route.
Careful analysis revealed some drawbacks but after correction the manuscript could be valuable paper which could be consider for publication in Molecules.
Below are some problems that could help Authors to improve their manuscript:
1. In the Introduction some information about imprinted polymers for zearalenone adsorption should be mentioned and examples should be added. It is important.
2. The novelty of the manuscript should be emphasise.
3. Not all abbreviations are included in list, e.g. MISG, CISG, ADs, MNPs etc.
4. Authors used different abbreviations for obtained products – it should be unified.
5. Section 2.1 should be written in better way, e.g. chemicals obtained from one company could be specified in one sentence, then the section will be shortened.
6. Magnetization measurement of Fe3O4@SiO2 could be added to show how addition of another layers impact on magnetization of the material.
7. Style of the text should be improved. The authors are also invited to refine the language of the paper to make the story flow smoothly. Grammatical mistakes should be corrected. Also, other editorial mistakes should be improved.
8. There is many mistakes according to figures citing in the text and numeration.
9. What was the aim of the study? I was to obtain selective sorbent for zearalenone or for the group of the analytes?
10. Figure 2 – the scale on the micrographs should be the same to compare obtained products.
In my opinion, the manuscript in current form is not suitable for publication. I recommend major revision of the manuscript.
Author Response
Thank you very much for your comments and we agree on many of them. The text has been thoroughly revised and corrected. We also apologize for the large number of minor errors. The writing style has also been corrected.
In this manuscript preparation and characterization process of magnetic core-shell molecularly imprinted nanoparticles for zearalenone determination was described. Obtained material was tested as solid phase sorbent during determination of analyte in water samples. The topic of the manuscript is interesting, especially in connection with interesting synthetic route.
Careful analysis revealed some drawbacks but after correction the manuscript could be valuable paper which could be consider for publication in Molecules.
Below are some problems that could help Authors to improve their manuscript:
- In the Introduction some information about imprinted polymers for zearalenone adsorption should be mentioned and examples should be added. It is important.
It has been included in the text.
- The novelty of the manuscript should be emphasise.
It has been included in the text.
- Not all abbreviations are included in list, e.g. MISG, CISG, ADs, MNPs etc.
Many of the abbreviations have been deleted and others have been explained.
- Authors used different abbreviations for obtained products – it should be unified.
It has been unified for increasing the clarity.
- Section 2.1 should be written in better way, e.g. chemicals obtained from one company could be specified in one sentence, then the section will be shortened.
This section has been totally changed.
- Magnetization measurement of Fe3O4@SiO2could be added to show how addition of another layers impact on magnetization of the material.
It is in the supl. material
- Style of the text should be improved. The authors are also invited to refine the language of the paper to make the story flow smoothly. Grammatical mistakes should be corrected. Also, other editorial mistakes should be improved.
The text has been thoroughly revised and corrected. The writing style has also been corrected.
- There is many mistakes according to figures citing in the text and numeration.
All them have been corrected.
- What was the aim of the study? I was to obtain selective sorbent for zearalenone or for the group of the analytes?
The work has focused in ZON because it has been found in waters. Nevertheless also the cross-reactivity has been checked, since the polymer is also able to recognize other metabolites and they could be also in these samples (but this fact has not been demonstrated yet)
- Figure 2 – the scale on the micrographs should be the same to compare obtained products.
All the scales have been corrected.
In my opinion, the manuscript in current form is not suitable for publication. I recommend major revision of the manuscript.
Round 2
Reviewer 2 Report
1. In line354,please check whether the MIPCIP is written correctly。
2. Please carefully check the use of words and interval symbols in the text, For example, spaces for symbols and wordsin line 533.
Author Response
- In line354,please check whether the MIPCIP is written correctly。
- Please carefully check the use of words and interval symbols in the text, For example, spaces for symbols and words in line 533.
Thank you for your comments. All these mistakes and others have been corrected.
Reviewer 3 Report
The authors have reflected the many review comments and the quality of the manuscript was gently improved. However, many errors are still presence in the manuscript. In my opinion, the authors should carefully revise the manuscript of current version for accepting this manuscript.
1) In Abstract, fluorescence (FLD) detection -> fluorescence detection (FLD)
2) P. 6, L. 190; P. 7, L. 300: ⁰C -> °C
3) P. 7, L. 296: MW..
4) P. 11, L. 434: Langmouir…
5) There are not presented a and b in Figure 6. Furthermore, the caption should be edited appropriately.
ex) Figure 6. (a) Equilibrium binding isotherms for ZON to the MIP (blue) and to the corresponding CIP (red) in phosphate buffer (50 mM, pH 7.5) and (b) Scatchard fitting for the MIP and the CIP. High affinity, blue lines. Low affinity, red lines.
6) The caption in Figure 7 should be also edited appropriately.
7) P. 15, L. 533: The world spacing is wrong.
Author Response
The authors have reflected the many review comments and the quality of the manuscript was gently improved. However, many errors are still presence in the manuscript. In my opinion, the authors should carefully revise the manuscript of current version for accepting this manuscript.
Thank you for your comments. The referee is right. All these mistakes and others have been corrected.
- In Abstract, fluorescence (FLD) detection -> fluorescence detection (FLD)
Corrected
2) P. 6, L. 190; P. 7, L. 300: ⁰C -> °C
Corrected
3) P. 7, L. 296: MW.
Corrected
4) P. 11, L. 434: Langmouir…
Corrected
5) There are not presented a and b in Figure 6. Furthermore, the caption should be edited appropriately.
Corrected
- ex) Figure 6. (a) Equilibrium binding isotherms for ZON to the MIP (blue) and to the corresponding CIP (red) in phosphate buffer (50 mM, pH 7.5) and (b) Scatchard fitting for the MIP and the CIP. High affinity, blue lines. Low affinity, red lines.
The figure caption has been changed.
6) The caption in Figure 7 should be also edited appropriately.
The figure caption has been changed.
7) P. 15, L. 533: The world spacing is wrong.
Corrected
Reviewer 4 Report
There are still some editorial and language mistakes in the manuscript.
E.g. line 243 - dot
line 245 - dioxane written with the use of capital letter
line 293 - two dots
In Materials section chemicals from Sigma Aldrich are mentioned in two sentences.
Author Response
There are still some editorial and language mistakes in the manuscript.
E.g. line 243 - dot
line 245 - dioxane written with the use of capital letter
line 293 - two dots
In Materials section chemicals from Sigma Aldrich are mentioned in two sentences.
Thank you for your comments. The referee is right. All these mistakes and others have been corrected.